# Irisin ameliorates myocardial ischemia-reperfusion injury by modulating gut microbiota and intestinal permeability in rats

Qingqing Liu[1,2‡], Yu Zhu[1,2,3‡], Guangyao Li[1,2], Tiantian Guo[1,2], Mengtong Jin[1,2], Duan Xi[1], Shuai Wang[1], Xuezhi Liu[1,2,4], Shuming Guo[1,2], Hui Liu[1,2,4], Jiamao Fan[1,2,5]*, Ronghua Liu[1,2]*

1 LinFen Central Hospital, LinFen, China, 2 Linfen Key Laboratory of Basic and Clinical Research on Coronary Heart Disease, Linfen Clinical Medical Research Center, LinFen, China, 3 School of Sport Medicine and Rehabilitation, Beijing Sport University, Beijing, China, 4 Department of Cardiovascular Surgery, Linfen Central Hospital, Linfen, China, 5 Department of Cardiology, Linfen Central Hospital, Linfen, China

‡ QL and YZ contributed equally to this work and share first authorship on this work.
* Fjmfjm0704@126.com (JF); 1311597497@qq.com (RL)

**Data Availability Statement:** The data that support the findings of this study are openly available in the BioSample database at http://www.ncbi.nlm.nih.

## Abstract

Recently, myocardial ischemia-reperfusion (I/R) injury was suggested associated with intestinal flora. However, irisin has demonstrated beneficial effects on myocardial I/R injury, thus increasing interest in exploring its mechanism. Therefore, whether irisin interferes in gut microbiota and gut mucosal barrier during myocardial I/R injury was investigated in the present study. Irisin was found to reduce the infiltration of inflammatory cells and fracture in myocardial tissue, myocardial enzyme levels, and the myocardial infarction (MI) area. In addition, the data showed that irisin reverses I/R-induced gut dysbiosis as indicated by the decreased abundance of Actinobacteriota and the increased abundance of Firmicutes, and maintains intestinal barrier integrity, reduces metabolic endotoxemia, and inhibits the production of proinflammatory cytokines interleukin 1β (IL-1β), interleukin 6 (IL-6), and tumor necrosis factor α (TNF-α). Based on the results, irisin could be a good candidate for ameliorating myocardial I/R injury and associated diseases by alleviating gut dysbiosis, endothelial dysfunction and anti-inflammatory properties.

## 1. Introduction

Acute myocardial infarction (AMI) is a common cardiac emergency associated with high rates of morbidity and mortality [1]. For patients with ST-elevation MI (STEMI), thrombolytic/fibrinolytic therapy or percutaneous coronary intervention (PCI) is considered the effective reperfusion strategy that should be performed [2]. Because reperfusion can induce excessive production of reactive oxygen species, excessive inflammatory response and cell apoptosis, the process is known as myocardial ischemia-reperfusion (I/R) injury [3]. Consequently, avoiding the occurrence of myocardial I/R injury is important for the treatment of ischemic heart

gov/bioproject/940075, reference number
[PRJNA940075].

**Funding:** This work was supported by the Scientific
research project of Shanxi Provincial Health
Commission (2022122) and the Linfen City key
research and development plan (2108).

**Competing interests:** The authors have declared
that no competing interests exist.

**Abbreviations:** *I/R*, ischemia reperfusion; *AMI*,
acute myocardial infarction; *STEMI*, ST elevation
myocardial infarction; *PCI*, percutaneous coronary
intervention; *SPF*, specific pathogen-fre; *ECG*,
electrocardiogram; *cTnI*, Cardiac Troponin I; *CK*,
Creatine phosphokinase; *LPS*, lipopolysaccharide;
*H&E*, hematoxylin and eosin; *IF*,
immunofluorescence; *ZO-1*, zonula occludens-1;
*DAPI*, 4' 6- diamidino- 2- phenylindol; *IL-1β*,
interleukin-1-beta; *IL-6*, interleukin-6; *TNF-α*,
tumour necrosis factor-alpha; *PCA*, Principal
component analysis; *PCoA*, principal coordinate
analysis; *PLS-DA*, partial least squares
discrimination analysis; *LDA*, linear discriminant
analysis; *LEfSe*, linear discriminant analysis effect
size; *ANOVA*, One-way analysis of variance.

disease. However, an effective therapy and potential target to prevent myocardial I/R injury in
patients does not yet exist.

Although the I/R mechanism is not yet clearly established, the gut microbiota appears to
play an important role in the development as shown in an increasing number of studies [4–6].
Vancomycin decreased heart's susceptibility to injury in an *in vivo* animal model of regional
myocardial I/R by reducing the abundance of gut microbiota [7]. In patients with STEMI, the
systolic function of the heart decreases, resulting in insufficient blood supply to systemic
organs including the intestine, leading to dysbiosis and changes in intestinal permeability [8].
However, the increase in intestinal permeability leads to the translocation of bacterial endotox-
ins into the blood, which contributes to a systemic inflammatory response [9]. Therefore, we
hypothesized that changes in gut microbiota and intestinal permeability were associated with
the occurrence of adverse cardiovascular events after myocardial I/R injury.

Irisin is a circulating hormone that is cleaved from the precursor protein fibronectin type
III domain containing 5 (FNDC5) [10]. Increasing evidence has shown that irisin has benefi-
cial effects on cardiovascular diseases [11–13]. Furthermore, in our previous studies, irisin
treatment was confirmed to modulate the mitochondrial function via the AMPK pathway, ulti-
mately protecting the H2C9 cardiomyocytes from hypoxia and reoxygenation injury [14]. In
addition, irisin showed potential to alleviate intestinal inflammation by altering the gut micro-
biota [15–17]. Irisin could also restore gut barrier function via the integrin αVβ5-AMPK-UCP
2 pathway [18]. Therefore, in the present study, the effects of irisin on intestinal bacteria and
intestinal barrier were evaluated in the rat model of myocardial I/R injury.

## 2. Materials and methods

### 2.1 I/R rat model establishment and treatment

Adult male Wistar rats weighing 230–250 g were purchased from the Laboratory Animal Cen-
ter of Shanxi Provincial People's Hospital (Taiyuan, China). The rats were housed under a 12 h
light/dark cycle at 23 ± 2˚C with 40%–60% humidity, and they had free access to standard
food and water. All animal experimental procedures were approved by the ethical committee
of LinFen Central Hospital (Permit Number: 2021-29-1).

After 1 week of adaptation, all rats were randomly divided into three groups, namely, a
sham-operated group (Sham, n = 10), a myocardial ischemia reperfusion injury group (I/R,
n = 10), and an irisin group (Irisin, n = 10). The rats in the Sham and I/R groups were intraper-
itoneally injected with 0.2 mL of PBS, while those in the Irisin group were intraperitoneally
injected with 0.2 mL of irisin (100 mg/kg, Sigma, USA, SRP8039) [19]. Treatment was contin-
ued for 7 days. The myocardial I/R injury model was induced in the Irisin and I/R groups
according to the previously described procedure (S1 Fig) [20]. The rats were intraperitoneally
injected with sodium pentobarbital (50 mg/kg) and intubated with a small-animal ventilator
(Shanghai Yuyan Instruments Co., Ltd., China) set at a respiratory rate of 60–70 breaths per
minute. The surgical area was disinfected, the left chest was opened at the third intercostal
space to expose the heart, and the pericardium was separated to exteriorize the heart. The left
anterior descending coronary artery was quickly ligated with 6.0 prolene suture for 30 min,
after which the suture was removed for reperfusion for 120 min. Meanwhile, the rats under-
went electrocardiography (ECG; Shanghai Yuyan Instruments Co., Ltd., China) with limb lead
II tracing during the operation. Previous studies revealed that ischemia for 30 min would lead
to significant elevation of the ST segment of ECG, and successful model establishment could
be indicated by a decrease of the ST segment by at least 50% following 120 min of reperfusion
(S2 Fig). Meanwhile, the sham operation included all procedures except ligation of the left
anterior descending coronary artery.

## 2.2 Histopathological and 2,3,5-triphenyl-2H-tetrazolium chloride (TTC) staining

The heart, colon, and ileum tissues were fixed in 4% buffered paraformaldehyde for 48 h and then embedded in paraffin. The sections (5 μm) were mounted on slides, deparaffinized in xylene, rehydrated in decreasing concentrations of ethanol, and subjected to hematoxylin and eosin (H&E) staining. The histological score was determined as previously described [21, 22].

To measure the area of MI, 2,3,5-triphenyl-2H-tetrazolium chloride (TTC) (Solarbio, China, T8170) staining was used. The left ventricle was cut transversely into six sections of the same thickness and stained with 2% TTC at 37°C for 30 min without exposure to light. After staining, normal areas of the myocardium were stained red and infarcted areas were left unstained.

## 2.3 TUNEL and immunofluorescence (IF) staining

The sections (5 μm) were deparaffinized in xylene and rehydrated in decreasing concentrations of ethanol. A TUNEL kit (Beyotime, China, C1088) was used for TUNEL staining according to the manufacturer's instructions. Apoptotic rate was counted as previously described [23]. The sections were submerged into Tris-ethylenediaminetetraacetic acid antigenic retrieval buffer and heated for 5 min by pressure cooker. The sections were then treated with 3% hydrogen peroxide in methanol, blocked with 5% bovine serum albumin, and incubated with anti-zonula occludens-1 (ZO-1, 1: 1000, Abcam, UK, ab221546) and anti-occludin antibodies (1: 200, Abcam, UK, ab216327) overnight at 4°C followed by incubation with Alexa Fluor 594-conjugated secondary antibodies (1: 500, Bioss, China, bs-0295G-AF594). Nuclear staining was performed using 4',6-diamidino-2-phenylindole (DAPI, BOSTER, China, AR1176). An Olympus inverted fluorescence microscope was used to observe section staining. The images were evaluated using ImageJ software.

## 2.4 Serum cardiac troponin I (cTnI), creatine phosphokinase (CK), lipopolysaccharide (LPS) and Zonulin measurements

Rats were sacrificed and their blood serum centrifuged. Serum levels of cardiac troponin I (cTnI), creatine phosphokinase (CK), lipopolysaccharide (LPS) and Zonulin were determined using the cTnI ELISA kit (Solarbio, China, SEKR-0048), CK ELISA kit (Solarbio, China, BC1145), LPS ELISA kit (Signalway Antibody, USA, EK3762) and Zonulin ELISA kit (Jianglaibio, China, JL45867) according to the manufacturer's instructions.

## 2.5 Western blot analysis

Colon and ileum tissues (0.5 g) were lysed on ice with 500 μL of RIPA lysis buffer for 30 min in the presence of protease and phosphatase inhibitors and then sonicated for 1 min at 60 Hz. After centrifugation at 12,000 rpm for 15 min at 4°C, the supernatant was harvested. Protein concentrations were determined with the BCA protein assay kit (Solarbio, China, PC0020) and the protein was thermally denatured at 100°C for 10 min. The protein was then isolated using sodium dodecyl sulfate polyacrylamide gels and transferred to nitrocellulose membranes. After blocking the membranes with skim milk powder, the membranes were incubated overnight at 4°C with rabbit anti-mouse primary antibodies including interleukin 1β (IL-1β, 1: 1000, Abcam, UK, ab283818), interleukin 6 (IL-6, 1: 1000, CST, USA, 12912T), tumor necrosis factor α (TNF-α, 1: 1000, CST, USA, 11948T), ZO-1 (1: 1000, Abcam, UK, ab221546); occludin (1: 1000, Abcam, UK, ab216327), and β-actin (1: 1000, CST, USA, 37000T). After washing with TBST, the membranes were incubated with goat anti-rabbit secondary antibody (1: 3000, CST, USA, 7074) at 37°C for 1 h followed by incubation with a chemiluminescent substrate for

visual detection using the Tanon imaging system. ImageJ software was used to calculate the integrated optical density (IOD).

## 2.6 Extraction of fecal DNA and high-throughput sequencing

The cecal contents of the rats were collected in sterile tubes and stored in a refrigerator at -80°C. DNA in the cecal contents was extracted using a DNA extraction kit (Accurate, China, 37000T) and the quality of DNA extraction was determined using 0.8% agarose gel electrophoresis and quantified with a UV spectrophotometer. The 16S rRNA hypervariable region (V3-V4) was amplified with PCR using the extracted DNA as a template. The primer sequences were `ACTGCATCCGCAGCGTCGA` and `CCTGTACGGTCTTGCATAT`. The PCR products were detected using 2% agarose gel electrophoresis and purified with AXYGEN gel extraction kit. The preliminary quantification results of electrophoresis were obtained by fluorescing the PCR-amplified products using the Quant-iT PicoGreen dsDNA assay kit (Thermo Fisher Scientific, USA, P7589). Based on the quantitative results, purified amplicons were pooled in equimolar amounts and paired-end sequenced on an Illumina MiSeq PE300 platform/Nova-Seq PE250 platform (Illumina, USA) according to Majorbio Bio-Pharm Technology Co. Ltd. (China) standard protocols.

## 2.7 Bioinformatic analysis

To minimize the effects of sequencing depth on α and β diversity measures, the number of 16S rRNA gene sequences from each sample were rarefied to 20,000, which still yielded an average Good's coverage of 99.09%. Bioinformatic analysis of the gut microbiota was performed using the Majorbio Cloud platform (https://cloud.majorbio.com). Based on the operation taxonomic unit (OTU) information, rarefaction curves and α diversity indices, including observed OTUs, Chao1 richness, Shannon index, and Good's coverage were calculated using Mothur v1.30.1. The similarity among the microbial communities in different samples was determined using principal component analysis (PCA), principal coordinate analysis (PCoA), and partial least squares discrimination analysis (PLS-DA) based on Bray–Curtis dissimilarity using Vegan v2.5–3 package. The PERMANOVA test was used to assess the percentage of variation explained by the treatment with its statistical significance using Vegan v2.5–3 package. The linear discriminant analysis (LDA) effect size (LEfSe; http://huttenhower.sph.harvard.edu/LEfSe) was performed to identify the significantly abundant taxa (phylum to genera) of bacteria among the different groups (LDA > 4.00, $p < 0.05$). Spearman correlation analyses were used to assess correlations between the 10 top genus and blood parameters, the infarct area, colon and ileum barrier function, degree of bacterial translocation, and inflammatory response.

## 2.8 Statistical analysis

The data are presented as the mean ± standard error of mean (SEM) of five or more independent experiments. Normal distribution was confirmed using the Shapiro–Wilk test. One-way analysis of variance (ANOVA), least significant difference (LSD), or Tamhane test were used to compare the statistical differences among multiple groups. A $p$-value $< 0.05$ was considered statistically significant. SPSS 22.0 was used for statistical analysis.

## 3. Results

### 3.1 Effects of irisin on gut microbial diversity and richness

To characterize the microbial populations in the rat gut, the bacterial populations were measured using 16S rRNA gene sequencing. The rarefaction curves of all samples indicated that

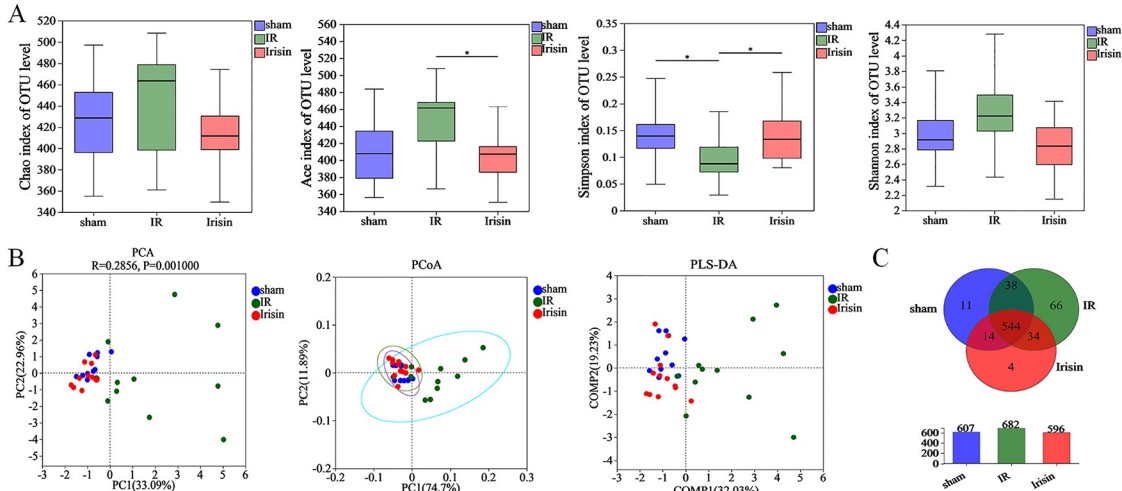

**Fig 1. Effects of irisin on gut microbial diversity and richness. A.** Effects of irisin treatment on α diversity was determined using Chao, Ace, Simpson, and Shannon indices; **B.** β diversity was. determined using Euclidean distance-based principal component analysis (PCA), weighted UniFrac distance-based principal coordinates analysis (PCoA), and partial least squares discrimination analysis (PLS-DA); **C.** Venn diagram illustrating overlap of operation taxonomic units (OTUs) in intestinal microbiota among the samples. Data are expressed as means ± standard error of the mean (SEM; n = 11 in each group). *$p < 0.05$; **$p < 0.01$; ***$p < 0.001$.

the sequencing coverage was sufficient to reflect the composition of intestinal flora (S3A Fig). Rank-Abundance curves showed that species were evenly distributed (S3B Fig). In gut microbial α diversity, both richness and evenness were indicated based on the Chao, ACE, Simpson, and Shannon indices. In I/R rats, ACE index and Shannon index increased and Simpson index decreased, and irisin completely restored these effects (Fig 1A).

Principal component analysis (PCA), principal coordinate analysis (PCoA), and partial least squares discrimination analysis (PLS-DA) were used to measure the difference between microbial communities. The aggregation of the flora in the I/R group significantly stayed away from Sham and Irisin groups, and the gut microbial community structure was similar between Sham and Irisin group (Fig 1B). Notably, the difference of the rat microbial community composition was small in the Irisin and the Sham groups.

Among 711 OTUs, 544 of the total richness were shared among all groups, and OTUs were observed between two groups or in each group (Fig 1C). In addition, irisin treatment decreased OTUs in the I/R rats.

These data indicated that irisin treatment significantly improved α and β diversity of intestinal microbiota.

### 3.2 Effects of irisin on the gut microbiota composition

Based on the results, gut microbiota was significantly changed. Therefore, the gut microbiota composition was compared among the three groups to identify potential probiotics or harmful bacteria of irisin intervention after I/R.

At the phylum level, I/R affected the relative abundance of Firmicutes and Actinobacteriota. Notably, irisin treatment significantly increased the relative abundance of Firmicutes and decreased the relative abundance of Actinobacteriota (Fig 2A).

The top 10 families were significantly affected by I/R; several families (*Corynebacteriaceae*, *Bifidobacteriaceae*, *Staphylococcaceae*, *Aerococcaceae*, *Akkermansiaceae*, and *Carnobacteriaceae*) were drastically increased and others (*Peptostreptococcaceae* and *Monoglobaceae*)

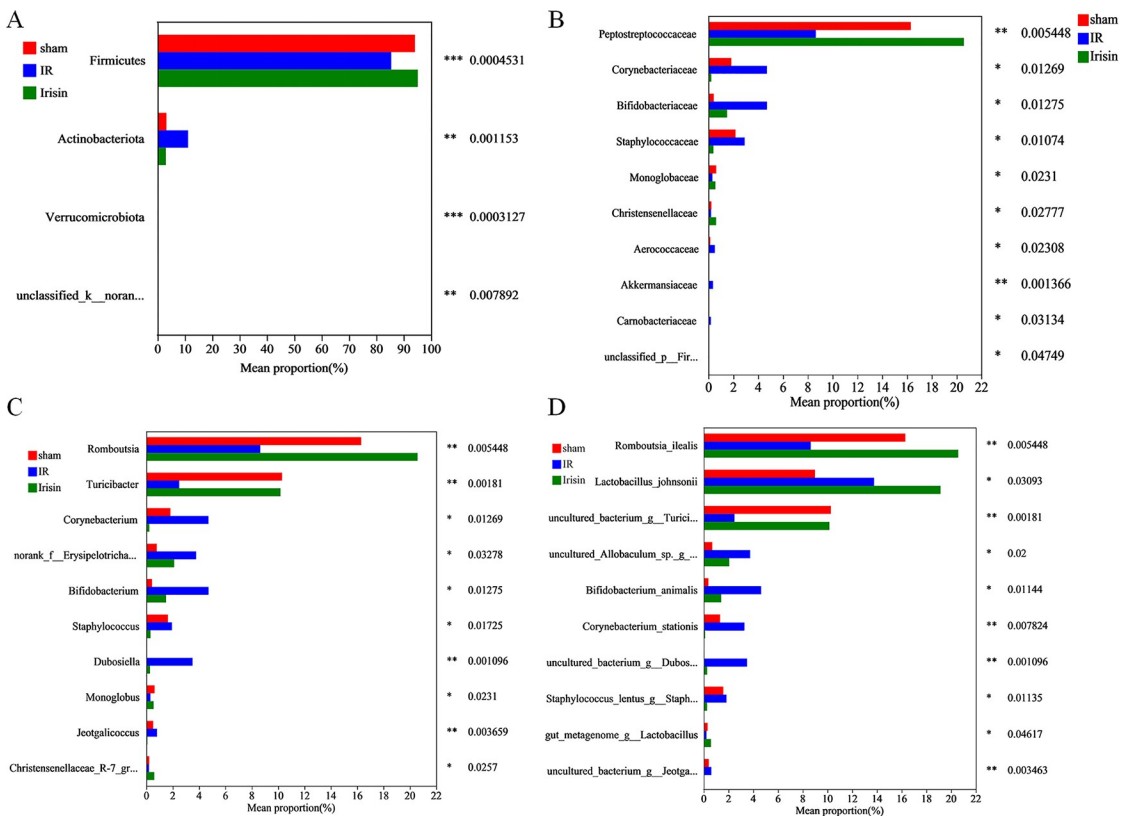

**Fig 2. Effects of irisin treatment on the gut microbiota composition. A.** Relative abundance of the major microbial phyla; **B.** Relative abundance of the top 10 different families; **C.** Relative abundance of the top 10 different genera; **D.** Relative abundance of the top 10 different species. Data are expressed as means ± standard error of the mean (SEM; n = 11 in each group).*$p < 0.05$; **$p < 0.01$; ***$p < 0.001$.

decreased. However, irisin significantly increased *Peptostreptococcaceae* and *Monoglobaceae* but decreased *Corynebacteriaceae*, *Bifidobacteriaceae*, *Staphylococcaceae*, *Aerococcaceae*, *Akkermansiaceae*, and *Carnobacteriaceae* to some extent (Fig 2B).

Next, the top 10 genera were analyzed. In the I/R group, several genera (*Corynebacterium*, *norank_f__Erysipelotrichaceae*, *Bifidobacterium*, *Staphylococcus*, and *Jeotgalicoccus*) were significantly increased, however, others (*Romboutsia*, *Turicibacter*, and *Monoglobus*) were markedly decreased. In contrast, irisin treatment significantly increased *Romboutsia*, *Turicibacter*, and *Monoglobus* but markedly reduced *Corynebacterium*, *norank_f__Erysipelotrichaceae*, *Bifidobacterium*, *Staphylococcus*, and *Jeotgalicoccus* (Fig 2C).

The relative abundance of 10 species was different in I/R-induced rats, with an increase of seven species (*Lactobacillus_johnsonii*, *uncultured_Allobaculum_sp._g__norank*, *Bifidobacterium_animalis*, *Corynebacterium_stationis*, *uncultured_bacterium_g__Dubosiella*, *Staphylococcus_lentus_g__Staphylococcus*, and *uncultured_bacterium_g__Jeotgalicoccus*) and a decrease of three species (*Romboutsia_ilealis*, *uncultured_bacterium_g__Turicibacter*, and *gut_metagenome_g__Lactobacillus*). In the irisin group, an increase of three species (*Romboutsia_ilealis*, *uncultured_bacterium_g__Turicibacter*, and *gut_metagenome_g__Lactobacillus*) and a decrease of six species (*uncultured_Allobaculum_sp._g__norank*, *Bifidobacterium_animalis*, *Corynebacterium_stationis*, *uncultured_bacterium_g__Dubosiella*, *Staphylococcus_lentus_g__- Staphylococcus*, and *uncultured_bacterium_g__Jeotgalicoccus*) was observed (Fig 2D).

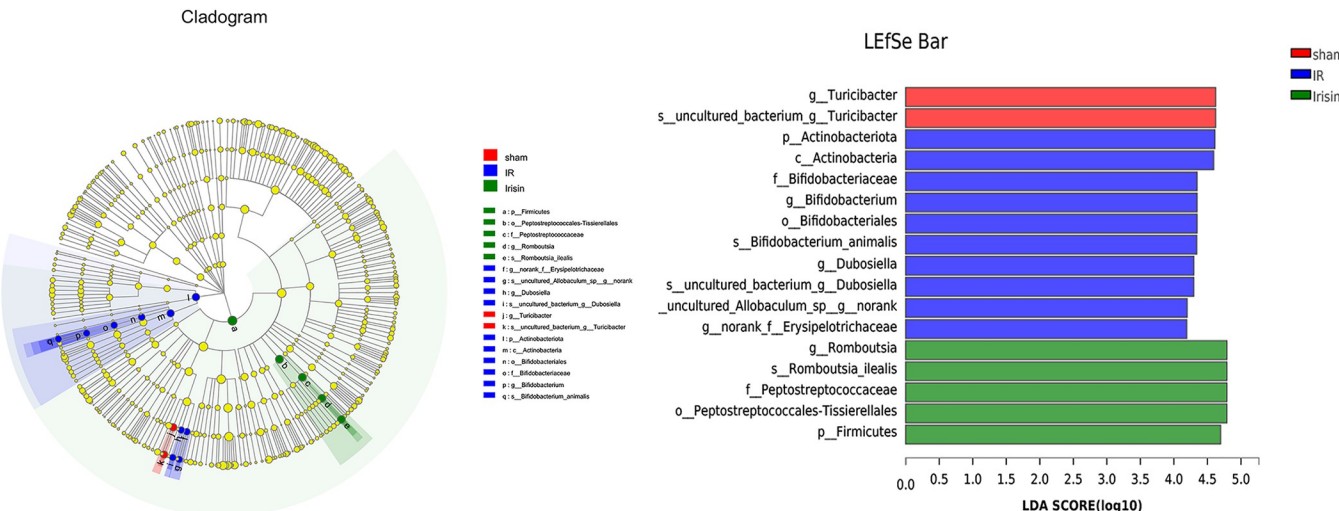

**Fig 3. Irisin alters gut microbiota biomarkers in ischemia-reperfusion (I/R) rats.** Identification of discriminant taxa among the four groups based on linear discrete analysis (LDA) effect size (LEfSe) analysis. Cladogram of the microbiota. Significant discriminant taxon nodes of the sham, I/R, and irisin groups are represented by red, blue, and green, respectively. Nondiscriminant taxon nodes are represented by yellow. The LDA score indicates the level of differentiation among the three groups. A threshold value of 4.0 was used as the cutoff level. Horizontal bar chart showing discriminant taxa. Significant discriminant taxa of the sham, I/R, and irisin groups are represented by red, blue, and green, respectively.

The bacterial composition with significant differences among the sham, I/R, and irisin groups was analyzed using LDA-LEfSe (Fig 3). In the sham group, *g__Turicibacter* and *s__uncultured_bacterium_g__Turicibacter* played critical roles and may be considered biomarkers. In addition, *p__Actinobacteriota*, *c__Actinobacteria*, *f__Bifidobacteriaceae*, *g__Bifidobacterium*, *o__Bifidobacteriales*, *s__Bifidobacterium_animalis*, *g__Dubosiella*, *s__uncultured_bacterium_g__Dubosiella*, *s__uncultured_Allobaculum_sp__g__norank*, and *g__norank_f__Erysipelotrichaceae* had an important function and may be used as biomarkers in the I/R group, and *g__Romboutsia*, *s__Romboutsia_ilealis*, *f__Peptostreptococcaceae*, *o__Peptostreptococcales-Tissierellales*, and *p__Firmicutes* played a crucial part and may be considered biomarkers in the irisin group.

### 3.3 Effects of irisin on the gut microbiota function phenotype

The gut flora performs basic physiological functions in the host and participates in regulating body's homeostasis and health. Therefore, BugBase was employed to predict the functional potential of bacteria. Then, from our research results, it was found that irisin treatment affected Biofilm Forming, Gram Positive, Gram Negative, Pathogenic Potential, Mobile Element Containing, Oxidative Stress Tolerant and Oxygen Utilizing (S4 Fig). Gram negative and Biofilm forming bacteria were more abundant in the I/R group. However, these effects were dramatically restored by irisin (S4 Fig). Concluding, irisin treatment improved gut microbiota function phenotype.

### 3.4 Irisin maintains intestinal integrity in I/R rats

The gut epithelial integrity is considered the first line of defense of the gastrointestinal tract. Intestinal dysbiosis in I/R animals may affect gut permeability and subsequently lead to release of potentially harmful bacterial metabolites into the blood [24]. In the current study, I/R dramatically increased intestinal permeability and damaged the intestinal mucosa (Figs 4 and 5) but were restored with irisin treatment.

The colon tissue of representative rats from each group showed the I/R group exhibited inflammatory cell infiltration (Fig 4A). Notably, the ileum villi arrangement was loose and

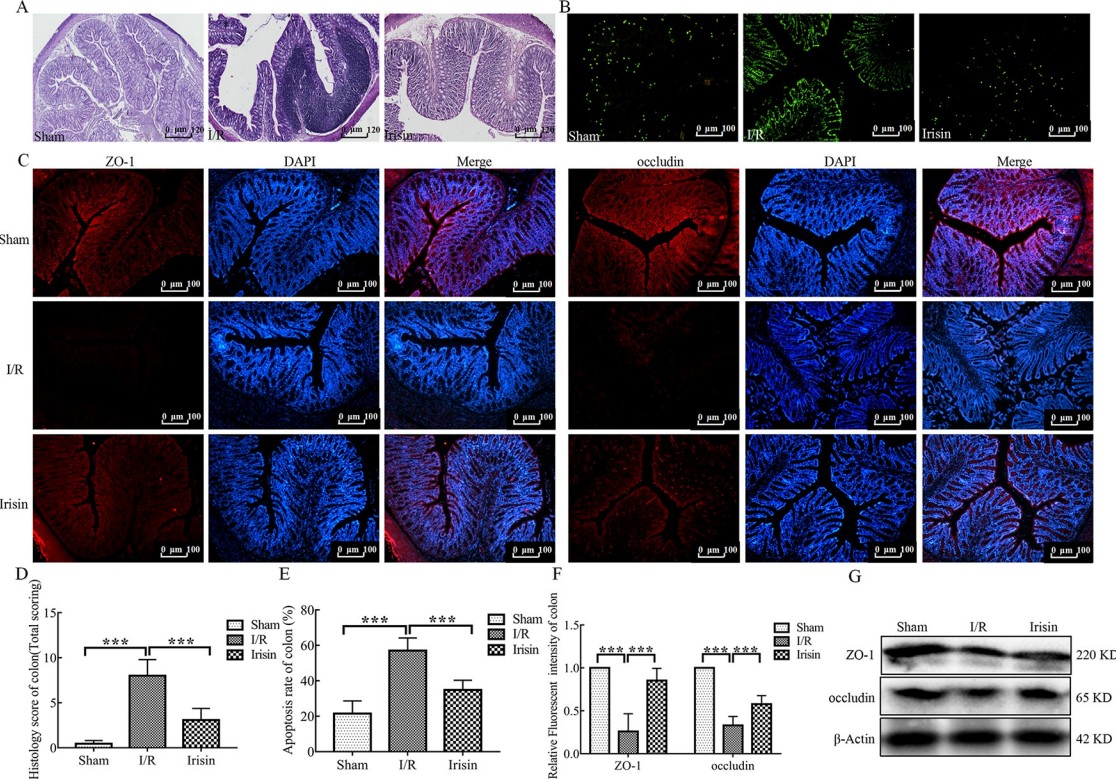

**Fig 4. Irisin maintains intestinal integrity in ischemia-reperfusion (I/R) rats. A.** Hematoxylin and eosin (H&E) staining of colon sections; **B.** Representative fluorescent pictures of TUNEL staining of colonic sections. The apoptotic cells were detected by TUNEL (green); **C.** Immunofluorescence staining of the infiltration of zonula occludens-1 (ZO-1, rad) and occludin (red) in colon tissue section; **D.** Histological scores for colon sections (n = 6); **E.** Apoptosis rate in colonic sections (n = 6); **F.** Quantification of staining intensity of ZO-1 and occludin in each group (n = 6); **G.** ZO-1 and occludin expression in the colon tissue from each group were evaluated using western blotting. Data are expressed as means ± standard error of the mean (SEM). *$p < 0.05$; **$p < 0.01$; ***$p < 0.001$.

disordered was observed in the I/R group (Fig 5A). In view of the obvious pathological changes of colon and ileum in I/R rats, the level of apoptosis was assessed by TUNEL staining. Compared to the Sham group, the levels of apoptosis of colon and ileum tissues were significantly increased in the I/R group (Figs 4B and 5B). After treatment with the irisin, the colon and the ileum structure and apoptosis level were alleviated. Inflammatory cell infiltration was decreased in colon tissue (Fig 4A) and the shape of ileum villi was straight finger-like protrusions neatly and densely arranged in the irisin group (Fig 5A).

Because intestinal mucosa damage is closely associated with the expression of epithelial mucosal proteins and tight junction proteins, the expression of ZO-1 and occludin was examined. Irisin significantly increased ZO-1 and occludin expression compared with the I/R group in both colon and ileum tissues (Figs 4C, 4G, 5C and 5G).

These findings indicate that irisin may enhance intestinal barrier integrity in myocardial I/R injury rats.

### 3.5 Effects of irisin on gut inflammation

Previous studies revealed that serum LPS and Zonulin level was positively correlated with intestinal permeability [24, 25]. The data obtained confirm that myocardial I/R significantly increased intestinal permeability, leading to release of bacterial LPS and Zonulin into the

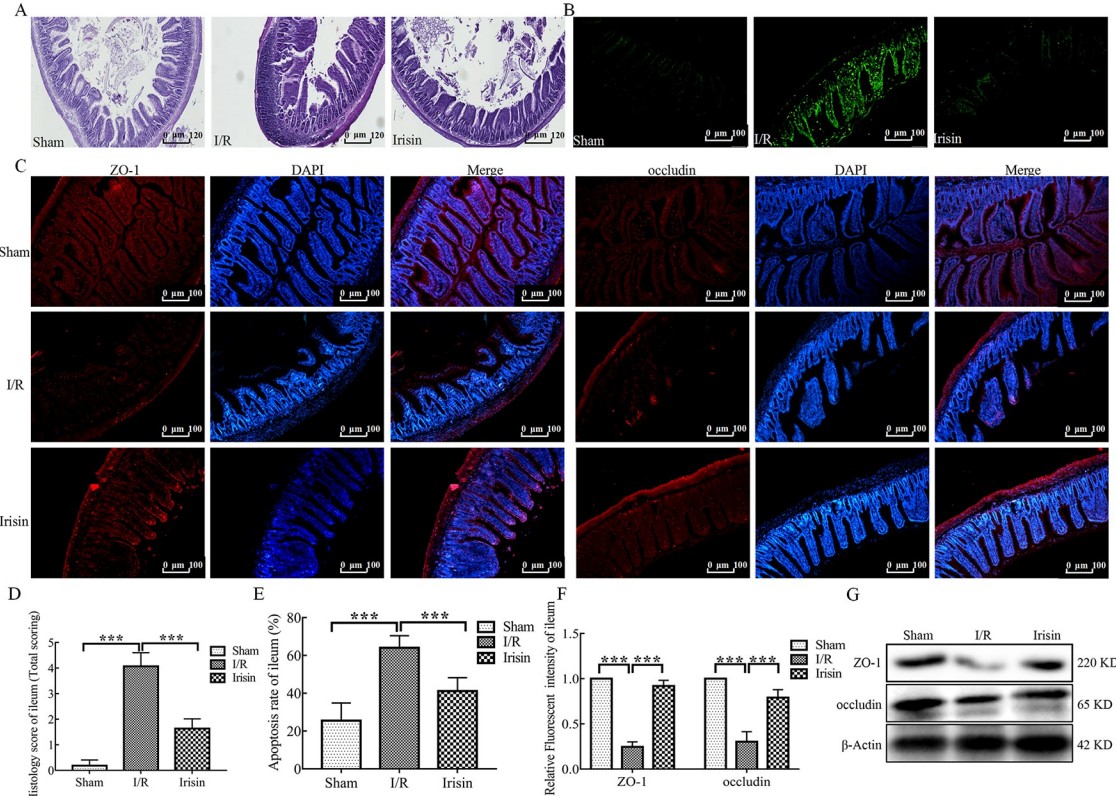

**Fig 5. Irisin maintains intestinal integrity in I/R rats. A.** Hematoxylin and eosin (H&E) staining of ileum sections; **B.** Representative fluorescent pictures of TUNEL staining of ileum sections. The apoptotic cells were detected by TUNEL (green); **C.** Immunofluorescence staining of the infiltration of zonula occludens-1 (ZO-1, red) and occludin (red) in ileum tissue sections; **D.** Histological scores for ileum sections (n = 6); **E.** Apoptosis rate in ileum sections (n = 6); **F.** Quantification of staining intensity of ZO-1 and occludin in each group (n = 6); **G.** ZO-1 and occludin expression in the ileum tissue from each group were evaluated using western blotting. Data are expressed as means ± standard error of the mean (SEM). $^*p < 0.05$; $^{**}p < 0.01$; $^{***}p < 0.001$.

blood, which was restored by irisin treatment (Fig 6A). Leaky gut was shown to produce higher levels of proinflammatory cytokines in colon or ileum tissues, including IL-1β, IL-6, and TNF-α [26]. In the present study, protein expression of these cytokines was measured using western blotting; IL-1β, IL-6, and TNF-α protein expression levels were higher in colon and ileum tissues of the I/R rats compared with the sham rats (Fig 6B and 6C). Notably, the protein expression level of these cytokines was altered by irisin treatment, resulting in the protein expression level similar to the sham rats (Fig 6B and 6C).

These results show irisin reduces inflammation in I/R rats by reducing proinflammatory cytokines in colon and ileum tissues.

## 3.6 Irisin reduces myocardial injury

In previous studies, ischemia for 30 min was shown to lead to significantly elevated ST segment of ECG, which decreased by at least 50% after 120 min reperfusion indicating successful model establishment [6]. The myocardial interstitium showed inflammatory cell infiltration and marked edema accompanied by dissolution, rupture and even necrosis of myocardial fibers in I/R rats [27]. In the current study, irisin treatment significantly reduced the elevated ST segment compared with I/R (S2 Fig). TTC staining was performed to analyze the infarct area. Compared with the I/R group, treatment with irisin notably reduced myocardial I/R-induced infarction (Fig 7A and 7D). In addition, the infiltration of inflammatory cells and

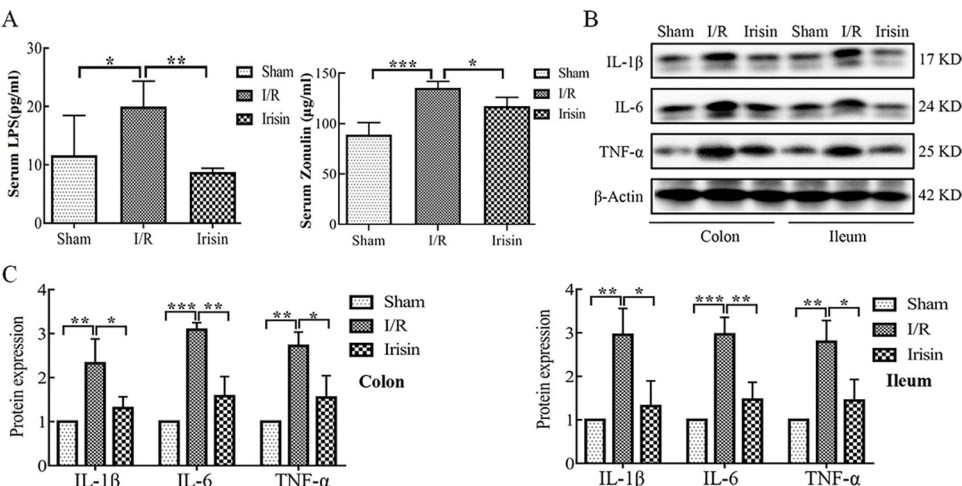

**Fig 6. Effects of irisin on gut inflammation. A.** Serum LPS and Zonulin concentration in each group detected using ELISA; **B.** Interleukin 1β (IL-1β), interleukin 6 (IL-6), and tumor necrosis factor α (TNF-α) expression in the colon and ileum tissue from each group were evaluated using western blotting; **C.** Quantification of relative protein expression of IL-1β, IL-6 and TNF-α. Data are expressed as means ± standard error of the mean (SEM). *$p < 0.05$; **$p < 0.01$; ***$p < 0.001$.

edema were reduced in the myocardial interstitium, and the myocardial fibers were intact and neatly arranged in the irisin group (Fig 7B). Fig 7C shows the apoptosis of cardiomyocytes was reduced in the irisin group. Cardiomyocyte necrosis can release a variety of myocardial enzymes to reflect the degree of myocardial injury. The levels of serum myocardial enzymes were increased in I/R group as shown in Fig 7E. Notably, the cTnI and CK levels were altered by irisin treatment, resulting in levels similar to sham rats.

The data obtained confirm that irisin ameliorates myocardial I/R injury.

## 3.7 Gut microbiota-associated blood parameters, the myocardial infarct area, colon and ileum barrier function, degree of bacterial translocation, and inflammatory response

Based on heatmap correlation analysis, in the 10 top genus, three genus were notably associated with blood parameters, the myocardial infarct area, colon and ileum barrier function, degree of bacterial translocation, and inflammatory response (Fig 8). In colon and ileum, *Turicibacter* was positively correlated with tight junction protein ZO-1 and occludin expression but negatively correlated with serum cTnI and CK levels, the myocardial infarct area, HE score, apoptosis rate, inflammatory factors (IL-1β, IL-6, and TNF-α) in colon and ileum and serum LPS and Zonulin concentration (Fig 8). Thus, these bacterial species may inhibit I/R development. In addition, *Bifidobacterium* and *norank_f__Erysipelotrichaceae* negatively correlated with tight junction protein ZO-1 and occludin expression in colon and ileum but positively correlated with serum cTnI and CK levels, the myocardial infarct area, HE score, apoptosis rate, inflammatory factors (IL-1β, IL-6 and TNF-α) in colon and ileum, and serum LPS and Zonulin concentration (Fig 8). Thus, these bacterial species may induce I/R. Collectively, these findings demonstrated the three bacterial species play vital roles in myocardial I/R injury.

## 4. Discussion

Evidence from a compilation of animal and human studies indicates the implications of gut microbiota and their metabolites in cardiovascular diseases are well established [28, 29]. In our

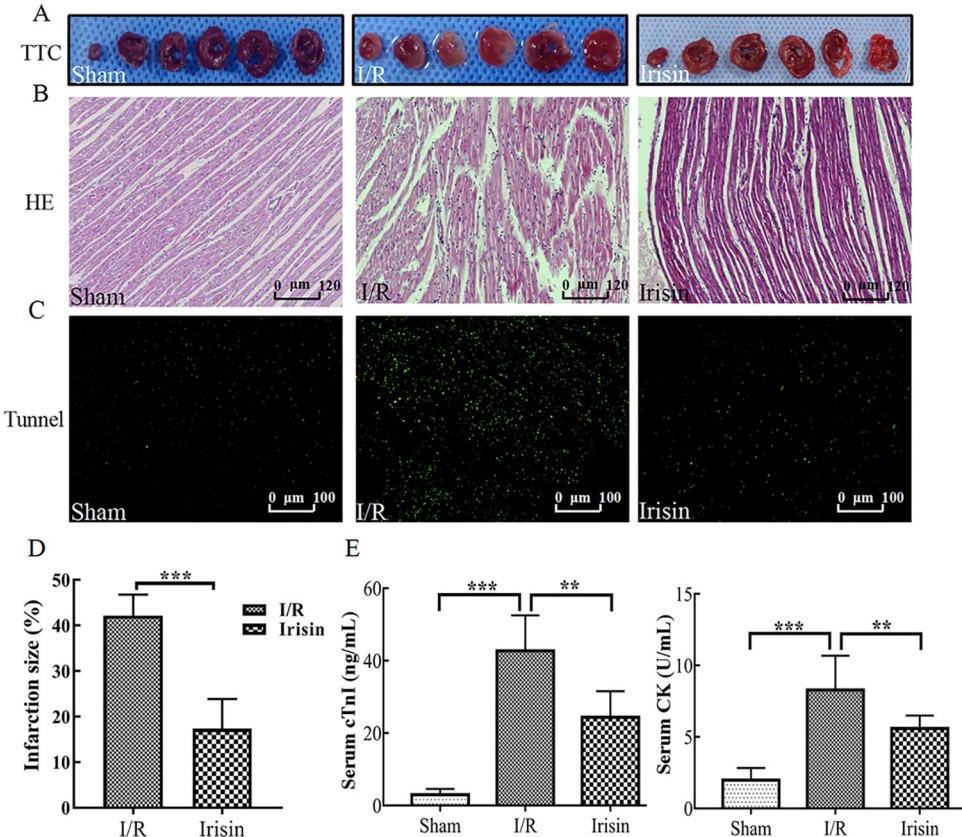

**Fig 7. Protective effects of irisin on myocardium in ischemia-reperfusion (I/R) rats. A.** 2,3,5-triphenyl-2H-tetrazolium chloride (TTC) staining of myocardium sections; **B.** Hematoxylin and eosin (H&E) staining of myocardium sections; **C.** Tunnel staining of myocardium sections; **D.** Quantification of the infarct area in TTC staining; **E.** Serum cardiac troponin I (cTnI) and creatine phosphokinase (CK) concentration in each group detected using ELISA. Data are expressed as means ± standard error of the mean (SEM). $^*p < 0.05$; $^{**}p < 0.01$; $^{***}p < 0.001$.

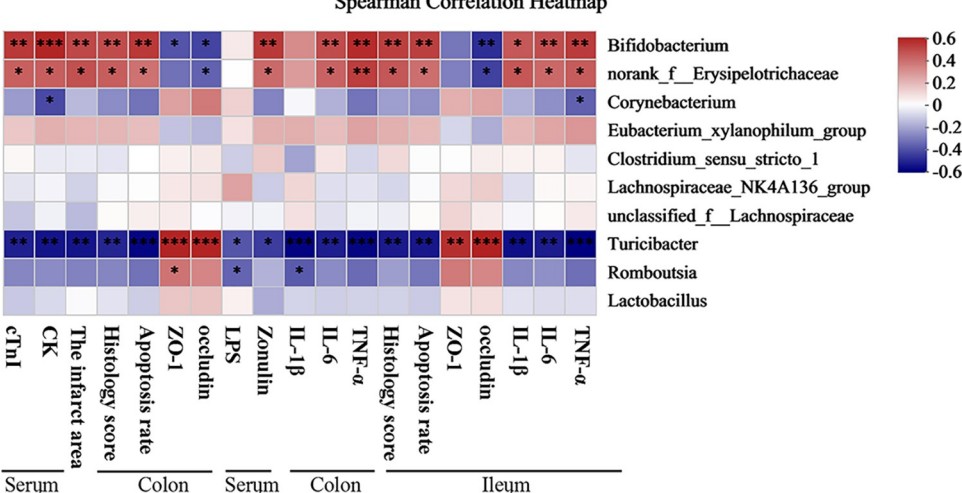

**Fig 8. The relationship among blood parameters, the infarct area, colon and ileum barrier function, degree of bacterial translocation and inflammatory response, and the 10 top genus estimated using Spearman correlation analysis.** $^*p < 0.05$; $^{**}p < 0.01$; $^{***}p < 0.001$.

previous studies, irisin was shown to protect the H2C9 cardiomyocytes from hypoxia and reoxygenation injury [14]. However, whether crosstalk exists between irisin, gut microbiota and cardioprotection remains unclear. In the present study, the data indicate that myocardial I/R injury is accompanied by intestinal microbiota imbalance. Notably, irisin treatment significantly decreased the abundance of gut microbiota. Furthermore, irisin treatment decreased gut inflammation and maintained the integrity of the intestinal barrier. To the best of our knowledge, this is the first study in which the effects of irisin on intestinal flora under myocardial I/R injury stress were investigated.

We performed 16S rRNA gene sequencing on the gut microbiota. In this study, significant microbiota changes in cecal contents in both I/R and irisin groups were observed. The results were compatible with a recent study in which α diversity reportedly significantly increased in I/R rats [30]. β diversity revealed significant separation of the community compositions between the sham and I/R groups, further confirming the possible close relationship between myocardial I/R injury and microbiota. Notably, the distribution of microbiota was not obvious in I/R rats, which may be due to the difference in the degree of acute stress. Furthermore, irisin prevented acute intestinal stress, thus, creating a microbiota similar to the sham group. Zhou *et al*. [8] suggested that increased richness and distinct structure of microbiome may be caused by the transportation of intestinal bacteria into the blood of I/R rats. In the present study, the relative abundance of gut bacteria significantly changed at the phylum, family, genera, and species level. LEfSe results were used to analyze the potential pathogenic bacteria such as *p__Actinobacteriota*, *c__Actinobacteria*, *f__Bifidobacteriaceae*, *g__Bifidobacterium*, *o__Bifidobacteriales*, *s__Bifidobacterium_animalis*, *g__Dubosiella*, *s__uncultured_bacterium_g__Dubosiella*, *s__uncultured_Allobaculum_sp__g__norank*, *g__norank_f__Erysipelotrichaceae* in the I/R rat intestine. *Bifidobacteriaceae* is a common probiotic that maintains gut microbiota balance [31]. However, the present study results are contradictory. If the relative abundance of probiotics is too high, the balance of bacterial flora may be disrupted and potentially become pathogenic bacteria. Data supporting the role of probiotics in other conditions are often conflicting [32]. A possible explanation for these results is that different gut environments may affect the abundance and composition of gut microbiota. Furthermore, *g__Romboutsia*, *s__Romboutsia_ilealis*, *f__Peptostreptococcaceae*, *o__Peptostreptococcales-Tissierellales*, and *p__Firmicutes* played a crucial part and could be considered a potential probiotic in the irisin group. *Romboutsia species*, such as *Romboutsia ilealis* [33] and *Peptostreptococcaceae* [34], could utilize glucose and carbohydrates to generate short-chain fatty acids to promote intestinal barrier integrity. Most Firmicutes are probiotics, except *Lactobacillus*, and other beneficial bacteria can produce butyrate [35]. In particular, butyrate is important for maintaining health by regulating the immune system and preserving the epithelial barrier [13, 36]. In summary, the findings showed the improvement of myocardial I/R injury due to irisin is closely associated with the changes in intestinal flora.

MI causes ischemic stress such as intestinal hypoperfusion, loss of tight junction protein occludin, intestinal mucosal damage, and increased intestinal permeability [37]. Tight junctions are important in the intestinal mucosal barrier and located at the top of the intestinal epithelium and consist of a number of proteins including ZO-1 and occludin of adjacent intestinal cells [38]. Zonulin is currently the only physiological regulator of intestinal permeability [39]. Upon stimulation by potentially harmful bacteria, Zonulin is released in large quantities into the intestinal lumen, where it binds to the Zonulin receptor. It can down-regulate the expression of tight junction protein, destroy the integrity of tight junction complex, and increase intestinal permeability [40]. In the current study, the structure of the colon and ileum was destroyed, the apoptosis of intestinal epithelial cells was significantly increased, the expression of ZO-1 and occludin was decreased and serum Zonulin levels were increased in

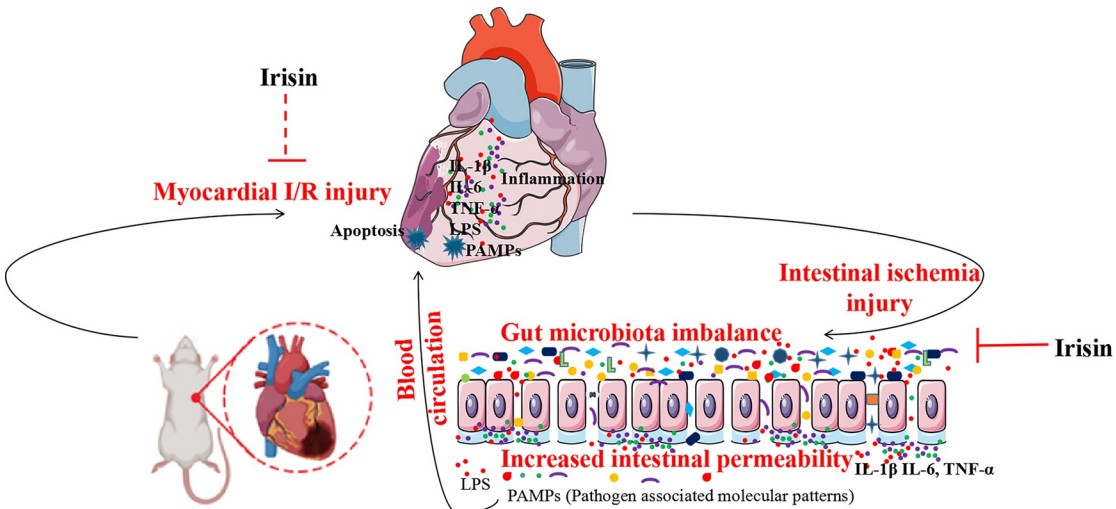

**Fig 9. Protective mechanisms of irisin against myocardial ischemia-reperfusion (I/R) injury.** Irisin supplementation restored the intestinal flora and intestinal barrier to inhibit bacteria and lipopolysaccharide (LPS) translocation as well as inhibit the inflammatory response and apoptosis of cardiomyocytes, thus, exerting cardioprotective effects.

the I/R rats. These results indicate that myocardial I/R can increase intestinal permeability in rats, however, irisin treatment can reverse this effect. Irisin possibly increases the abundance of beneficial bacteria in the intestine to maintain intestinal barrier integrity. Gut barrier breakdown leads to bacteria and endotoxin (LPS) translocation into the systemic circulation [41]. Then, low levels of LPS in the blood may activate TLR4 signaling in various cells, causing systemic and targeted inflammation [42–44]. In the present study, serum LPS concentration and the protein expression of inflammatory factors in colon and ileum tissues were determined. The results indicate that irisin inhibited the transfer of LPS from the intestine to the systemic circulation and caused overexpression of inflammatory factors IL-1β, IL-6, and TNF-α in I/R rats. Therefore, the beneficial effects induced by irisin treatment may be attributed to specific changes in the gut microbiota and maintenance of intestinal barrier integrity.

Furthermore, our data indicated that *Turicibacter* was obviously and negatively correlated with serum cTnI and CK levels, the myocardial infarct area, HE score, apoptosis rate, inflammatory factors (IL-1β, IL-6, and TNF-α) in colon and ileum and serum LPS and Zonulin concentration. Tang et al. found that the gut microbiota was a crucial element necessary for optimal cardiac repair after myocardial infarction (MI). Gut microbiota-derived short-chain fatty acids (SCFAs) alleviate the inflammatory microenvironment after myocardial infarction by regulating the immune system [45]. SCFAs could not only maintain intestinal barrier function, but also regulate immune response to inhibit inflammation [46]. *Turicibacter* is classified as beneficial bacteria that promote the production of SCFAs [47]. *Turicibacter* was significantly increased in the Irisin group. In this case, our experiments provided unequivocal evidence for protective role of irisin in myocardial I/R injury and irisin or probiotics supplementation may be an alternative or adjunct therapy for cardiovascular diseases treatment.

The results of this study indicated that intestinal microbiota is involved in irisin alleviating myocardial I/R injury. The effects of gut microbiota and intestinal barrier are likely a part of the mechanism underlying the irisin treatment process. However, the pathway through which irisin affects intestinal microbiota and its effects on metabolites need to be further investigated.

## 5. Conclusion

Myocardial I/R promotes alterations in gut microbiota that may enhance intestinal permeability and LPS translocation. However, irisin supplementation reversed the changes in gut microbiota, restored intestinal barrier structure, reduced LPS translocation, and decreased the inflammatory response, thus, exerting cardioprotective effects. In conclusion, irisin treatment can be an important tool for preventing and treating patients with myocardial I/R and intestinal flora is the mechanism affecting myocardial I/R injury (Fig 9). The results can be used to develop new prevention and diagnostic strategies to promote the health of patients with myocardial I/R.

## Supporting information

**S1 Fig. Experimental design.** Timeline of the experimental process of this research treatment and ischemia-reperfusion (I/R) injury induction in rats. (i. p.: intraperitoneal injection). (TIF)

**S2 Fig. The effect of irisin on the electrocardiogram waveform was evaluated in I/R rats.** (TIF)

**S3 Fig. Sample sequencing evaluation.** (A) Rarefaction curve indicating that the amount of sequencing reads per sample has reached saturation. (B) Rank-Abundance curves were shown. (TIF)

**S4 Fig. The gut microbiota function phenotype characterization of different groups based on BugBase analysis.** The relative abundance of Biofilm Forming, Gram Positive, Gram Negative, Pathogenic Potential, Mobile Element Containing, Oxidative Stress Tolerant and Oxygen Utilizing were shown. (TIF)

## Author Contributions

**Conceptualization:** Qingqing Liu, Yu Zhu, Xuezhi Liu, Shuming Guo, Jiamao Fan, Ronghua Liu.

**Data curation:** Qingqing Liu, Yu Zhu, Guangyao Li, Shuai Wang, Jiamao Fan.

**Formal analysis:** Qingqing Liu, Yu Zhu, Jiamao Fan.

**Funding acquisition:** Jiamao Fan.

**Investigation:** Mengtong Jin, Jiamao Fan.

**Methodology:** Tiantian Guo, Mengtong Jin, Duan Xi, Jiamao Fan.

**Project administration:** Xuezhi Liu, Ronghua Liu.

**Supervision:** Shuming Guo, Hui Liu.

**Validation:** Duan Xi, Shuming Guo, Ronghua Liu.

**Writing – original draft:** Qingqing Liu, Duan Xi.

**Writing – review & editing:** Qingqing Liu, Yu Zhu, Jiamao Fan.

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
