## [Decision Letter · Decision Letter 0]

2 May 2023

PONE-D-23-09095Irisin ameliorates myocardial ischemia-reperfusion injury by modulating gut microbiota and intestinal permeability in ratsPLOS ONE

Dear Dr. Fan,

Thank you for submitting your manuscript to PLOS ONE. After careful consideration, we feel that it has merit but does not fully meet PLOS ONE’s publication criteria as it currently stands. Therefore, we invite you to submit a revised version of the manuscript that addresses the points raised during the review process.

Several major concerns were raised by the reviewers.  It was felt that while the manuscript has merit, the barrier integrity and tight junction protein data needs to be further strengthened by additional experiments suggested.  Also, the presentation and the language of the manuscript needs significant improvement as suggested by the reviewers.  Please address each concern of the reviewer in a point by point manner.

We look forward to receiving your revised manuscript.

Kind regards,

Pradeep Dudeja

Academic Editor

PLOS ONE

Journal Requirements:

“YES - Specify the role(s) played.”

Reviewers' comments:

Reviewer's Responses to Questions

**Comments to the Author**

1. Is the manuscript technically sound, and do the data support the conclusions?

Reviewer #1: Partly

Reviewer #2: Yes

2. Has the statistical analysis been performed appropriately and rigorously? 

Reviewer #1: Yes

Reviewer #2: Yes

3. Have the authors made all data underlying the findings in their manuscript fully available?

Reviewer #1: Yes

Reviewer #2: Yes

4. Is the manuscript presented in an intelligible fashion and written in standard English?

Reviewer #1: Yes

Reviewer #2: Yes

5. Review Comments to the Author

Reviewer #1: The manuscript presents interesting data and shows the beneficial effects of Irisin in terms of reversing the I/R induced gut dysbiosis, its role in maintaining barrier integrity and its inhibitory effects on production of pro-inflammatory cytokines. Although the experiments were well designed and conducted but certain concerns were noted which significantly reduce the enthusiasm. The following concerns outlined below need to be addressed to improve the overall impact of the work.

Major Concerns:

1. Authors should provide detailed flowchart or diagrammatic representation of rat model establishment and treatment in section 2.1. It is not clear whether the I/R injury was induced in the Irisin treated group or not.

2. In section 2.3 (line 98), authors have mentioned that the sections were submerged into antigen retrieval buffer and heated for 5 min at 121°C in an autoclave. Please elaborate how this heating procedure was done in an autoclave?

3. In section 3.1, PCA, PCoA and PLS-DA plots should be elaborated in more detail.

4. In section 3.4, the results of TUNEL staining should be explained in a detailed manner. The legends for figure 4 should be more comprehensively written.

5. In section 3.4, the expression of ZO-1 and occludin is examined by immunostaining only. Authors should analyze the expression of tight junction proteins at RNA as well as protein levels by real time PCR and western blotting.

6. Authors should show higher magnification (100x) for occluding and ZO1 as the expression of both the proteins does not look like punctated as reported by other studies and recalculate the relative fluorescence intensity for the same. The number of images used for calculating the relative fluorescence intensity should be mentioned in the figure legend.

7. In figure 4 (A, B and C), Figure 5 (A, B and C) and Figure 7 (B and C) scale bars are not clearly visible on the images. The legends for these figures should be written in a more detailed and clear manner.

8. In section 3.6 and 3.7, the results of TTC staining and heat map of correlation analysis should be explained in more detail. Authors should provide references of other studies that have shown the correlation of bacterial species having impact on inhibition or development of I/R or other clinical conditions for supporting their results.

9. In lines 281- 282, authors have mentioned that significant microbiota changes in cecal contents in both I/R and irisin groups were observed. Please provide more details in section 2.6, whether these changes were analyzed in fecal or cecal content.

10. The authors should follow the same format for references, as there are multiple discrepancies in references. DOI for first 22 references are not mentioned in the manuscript.

11. The discussion section of the manuscript should emphasize more on highlighting the key findings of the study in a more extensive manner as well as providing more references for supporting the results.

Minor Concerns:

1. The manuscript needs to be read by a native English speaker as there are number of grammatical errors and punctuation mistakes. The manuscript should be proofread carefully.

2. In line 138, abbreviation of operation taxonomic unit (OTU) is written incorrectly as OUT.

3. In lines 184 and 188, repetition of “Staphylococcus” is observed.

4. Authors should provide the details about the working dilutions of antibodies, catalogue numbers of all the antibodies and kits used in the study.

5. The image quality for S3 Fig. should be improved.

6. The legends for figures 4 and 5 should be written in alphabetical order.

Reviewer #2: The manuscript "Irisin ameliorates myocardial ischemia-reperfusion injury by modulating gut microbiota and intestinal permeability in rats" was well written. The authors designed the appropriate experiments and classically presented the data. Much research documented Irisin and its importance in health diseases well. The authors utilized the available evidence and linked how irisin ameliorated myocardial I/R-mediated gut bacterial dysbiosis and barrier integrity. However, the authors need minor corrections and rewrite some of the result presentations and may need some additional experiments, which will help the manuscript strong for publication so the readers can understand clearly.

1. In the methodology section:

2.1 I/R rat model establishment and treatment. Line 74. The authors mentioned that based on reference 12, they made I/R injuries to the rats. Unfortunately, that reference did not explain the methodologies of creating myocardial I/R injury in rats. The authors requested to refer to the appropriate validated methodology reference.

2. 2.8 Statistical analysis: Line 149. The authors mentioned the data are presented as the mean±standard deviation (SD) in the results. However, none of the results and the figures are displayed with SD instead of Standard Error mean. The authors are requested to change SD to SEM in the statistical analysis section to avoid confusing readers.

3. In the Result section:

From lines 183 to 207: The authors explained that bacterial strains are differentially expressed in the I/R model and reverse in Irisin-treated models. However, the authors repeated the bacterial names, which were hard to read and understand, and reviewed the results. The authors are requested to simplify the presentation in an easily understandable manner.

4. Figure legend section:

Lines 494-512: Figure 4, Figure 5, and Figure 6: The authors did not explain the subsections' (A-F) self-explanatory information in the legends alphabetically. Comparing what is shown in the figures and the figure legends is hard. The authors are requested either change the subsections alphabetically based on the figure legends or figures accordingly.

5. Additional experiments may need to more strongly validate the Irisin mediated reversal of barrier integrity findings. The authors used only LPS levels for blood parameters to validate the barrier dysfunction. The authors are requested to consider to do to measure Zonulin and FITC-Dextran levels in the blood in both experimental models, which will strengthen the manuscript.

6. PLOS authors have the option to publish the peer review history of their article (what does this mean?). If published, this will include your full peer review and any attached files.

Reviewer #1: No

Reviewer #2: No

---

## [Author Response · Author response to Decision Letter 0]

31 Jul 2023

1. Authors should provide detailed flowchart or diagrammatic representation of rat model establishment and treatment in section 2.1. It is not clear whether the I/R injury was induced in the Irisin treated group or not.

Response: Thanks for your suggestion. We have added the detailed flowchart of rat model establishment and treatment in Supporting Information in the revised manuscript.

The detailed flowchart are shown as follows.

S1 Fig. Experimental design. Timeline of the experimental process of this research treatment and ischemia-reperfusion (I/R) injury induction in rats. (i. p.: intraperitoneal injection)

2. In section 2.3 (line 98), authors have mentioned that the sections were submerged into antigen retrieval buffer and heated for 5 min at 121°C in an autoclave. Please elaborate how this heating procedure was done in an autoclave?

Response: Thank you, the mistake has been corrected. We have revised the sentence as “The sections were submerged into Tris-ethylenediaminetetraacetic acid antigenic retrieval buffer and heated for 5 min by pressure cooker.” in the revised manuscript and highlighted it in red colour. 

The sections (5 μm) were deparaffinized in xylene and rehydrated in decreasing concentrations of ethanol. The Pressure cooker containing the tris-ethylenediaminetetraacetic acid antigenic retrieval buffer was heated to a boil on an electromagnetic stove. The sections were placed in antigen repair solution and capped for heat treatment, which ended 5 minutes after air injection from the pressure cooker valve. Then, the sections were naturally cooled. In general, high-pressure instruments can be heated to a temperature of about 120°C.

3. In section 3.1, PCA, PCoA and PLS-DA plots should be elaborated in more detail.

Response: Thanks for your suggestion. We have revised the “PCA, PCoA, and PLS-DA were used to measure the degree of difference between microbial communities. After treatment with irisin, aggregation of flora was not observed in the I/R rats (Fig 1B).” as “Principal component analysis (PCA), principal coordinate analysis (PCoA), and partial least squares discrimination analysis (PLS-DA) were used to measure the difference between microbial communities. The aggregation of the flora in the I/R group significantly stayed away from Sham and Irisin groups, and the gut microbial community structure was similar between Sham and Irisin group (Fig 1B).” 

4.In section 3.4, the results of TUNEL staining should be explained in a detailed manner. 

Response: Thanks for your suggestion. We have added more details in section 3.4 and highlighted it in red colour. The legends for figure 4 have been comprehensively written.

5. In section 3.4, the expression of ZO-1 and occludin is examined by immunostaining only. Authors should analyze the expression of tight junction proteins at RNA as well as protein levels by real time PCR and western blotting.

Response: Thanks for your suggestion. According to your suggestion, we have supplemented the some experimental results in Fig 4 and Fig 5 in the revised manuscript. We found that there is no statistical difference in the expression of ZO-1 and occludin genes in the colon and ileum by real time PCR. Nevertheless, protein expression of ZO-1 and occludin was increased in the Irisin group by immunofluorescence and Western Blotting. We analyzed that irisin might enhance the expression of ZO-1 and occludin proteins through acting on transcription or translation processes.

6. Authors should show higher magnification (100x) for occluding and ZO1 as the expression of both the proteins does not look like punctated as reported by other studies and recalculate the relative fluorescence intensity for the same. The number of images used for calculating the relative fluorescence intensity should be mentioned in the figure legend.

Response: Thanks for your suggestion. Our team is relatively mature in the detection of ZO-1 and occludin protein expression and does not show punctate distribution [1-2]. The expression of ZO-1 and occludin proteins showed a zonal distribution. Figures of ZO-1 and occludin protein expression in this manuscript showed a magnification of 100×. Scale bars were added clearly on the images. Meanwhile, the number of images used for calculating the relative fluorescence intensity have be mentioned in the revised manuscript.

[1] Liu Q, Yang H, Kang X, et al. A Synbiotic Ameliorates Con A-Induced Autoimmune Hepatitis in Mice through Modulation of Gut Microbiota and Immune Imbalance. Mol Nutr Food Res. 2023 Apr;67(7):e2200428. doi: 10.1002/mnfr.202200428.

[2] Liu Q, Tian H, Kang Y, et al. Probiotics alleviate autoimmune hepatitis in mice through modulation of gut microbiota and intestinal permeability. J Nutr Biochem. 2021 Dec;98:108863. doi: 10.1016/j.jnutbio.2021.108863.

7. In figure 4 (A, B and C), Figure 5 (A, B and C) and Figure 7 (B and C) scale bars are not clearly visible on the images. The legends for these figures should be written in a more detailed and clear manner.

Response: Thanks for your suggestion. Scale bars have been added clearly on the images. 

8. In section 3.6 and 3.7, the results of TTC staining and heat map of correlation analysis should be explained in more detail. Authors should provide references of other studies that have shown the correlation of bacterial species having impact on inhibition or development of I/R or other clinical conditions for supporting their results.

Response: Thanks for your suggestion. We have added more details in Discussion and highlighted it in red colour.

We have performed a correlation analysis between TTC staining results and the 10 top genus (Fig. 8). Our data indicated that Turicibacter was obviously and negatively correlated with serum cTnI and CK levels, the myocardial infarct area, HE score, apoptosis rate, inflammatory factors (IL-1β, IL-6, and TNF-α) in colon and ileum and serum LPS and Zonulin concentration. Tang et al. found that the gut microbiota is a crucial element necessary for optimal cardiac repair after myocardial infarction (MI). Gut microbiota-derived short-chain fatty acids (SCFAs) alleviate the inflammatory microenvironment after myocardial infarction by regulating the immune system [1]. SCFAs could not only maintain intestinal barrier function, but also regulate immune response to inhibit inflammation [2]. Turicibacter is classified as beneficial bacteria that promote the production of SCFAs [3]. Turicibacter was significantly increased in the Irisin group. In this case, our experiments provided unequivocal evidence for protective role of irisin in myocardial I/R injury and irisin or probiotics supplementation may be an alternative or adjunct therapy for cardiovascular diseases treatment.

[1]Tang TWH, Chen HC, Chen CY, et al. Loss of Gut Microbiota Alters Immune System Composition and Cripples Postinfarction Cardiac Repair. Circulation. 2019 Jan 29;139(5):647-659. doi: 10.1161/CIRCULATIONAHA.

[2]Parada Venegas D, De la Fuente MK, Landskron G, et al. Short Chain Fatty Acids (SCFAs)-Mediated Gut Epithelial and Immune Regulation and Its Relevance for Inflammatory Bowel Diseases. Front Immunol. 2019 Mar 11;10:277. doi: 10.3389/fimmu.2019.00277.

[3]Liu X, Zhang Y, Li W, et al. Fucoidan Ameliorated Dextran Sulfate Sodium-Induced Ulcerative Colitis by Modulating Gut Microbiota and Bile Acid Metabolism. J Agric Food Chem. 2022 Nov 30;70(47):14864-14876. doi: 10.1021/acs.jafc.2c06417.

Fig 8. The relationship among blood parameters, the infarct area,colon and ileum barrier function, degree of bacterial translocation and inflammatory response, and the 10 top genus estimated using Spearman correlation analysis. *p < 0.05; **p < 0.01; ***p < 0.001

9. In lines 281- 282, authors have mentioned that significant microbiota changes in cecal contents in both I/R and irisin groups were observed. Please provide more details in section 2.6, whether these changes were analyzed in fecal or cecal content.

Response: Thanks for your suggestion. We have added more details in section 2.6 and highlighted it in red colour.

10. The authors should follow the same form at for references, as there are multiple discrepancies in references. DOI for first 22 references are not mentioned in the manuscript.

Response: We have adapted the format of the references to be consistent in the revised manuscript.

11. The discussion section of the manuscript should emphasize more on highlighting the key findings of the study in a more extensive manner as well as providing more references for supporting the results.

Response: Thanks for your suggestion. We have added more details in Discussion and highlighted it in red colour.

Minor Concerns:

1. The manuscript needs to be read by a native English speaker as there are number of grammatical errors and punctuation mistakes. The manuscript should be proofread carefully.

Response: Thanks for your suggestion. Before the first submission, we had finished polishing the language. We again thoroughly checked for the usage of the English language, typos and syntax and grammar errors carefully, and all the changes have been highlighted in red. 

2. In line 138, abbreviation of operation taxonomic unit (OTU) is written incorrectly as OUT.

Response: Thank you for your suggestion, OUT has been revised in the revised manuscript.

3. In lines 184 and 188, repetition of “Staphylococcus” is observed.

Response: Thank you for your suggestion, repeated words have been deleted in the revised manuscript. 

4. Authors should provide the details about the working dilutions of antibodies, catalogue numbers of all the antibodies and kits used in the study.

Response: Thanks for your suggestion. We have added the details about the working dilutions of antibodies, catalogue numbers of all the antibodies and kits in red colour.

5. The image quality for S3 Fig. should be improved.S3

Response: Thanks for your suggestion. We have revised S3 Fig in the revised manuscript, and show it as follows. 

S3 Fig. The effect of irisin on the electrocardiogram waveform was evaluated in I/R rats.

6. The legends for figures 4 and 5 should be written in alphabetical order.

Response: Thanks for your suggestion. The legend for figures 4 and 5 have been corrected in the revised manuscript.

Reviewer #2: The manuscript "Irisin ameliorates myocardial ischemia-reperfusion injury by modulating gut microbiota and intestinal permeability in rats" was well written. The authors designed the appropriate experiments and classically presented the data. Much research documented Irisin and its importance in health diseases well. The authors utilized the available evidence and linked how irisin ameliorated myocardial I/R-mediated gut bacterial dysbiosis and barrier integrity. However, the authors need minor corrections and rewrite some of the result presentations and may need some additional experiments, which will help the manuscript strong for publication so the readers can understand clearly.

1. In the methodology section:

2.1 I/R rat model establishment and treatment. Line 74. The authors mentioned that based on reference 12, they made I/R injuries to the rats. Unfortunately, that reference did not explain the methodologies of creating myocardial I/R injury in rats. The authors requested to refer to the appropriate validated methodology reference.

Response: Thanks for your suggestion. We have added the appropriate validated methodology reference in the revised manuscript. “The myocardial I/R injury model was induced in the Irisin and I/R groups according to the previously described procedure (Supplementary S1 Fig.) [20].”

[20] Zhao W, Wu Y, Ye F, et al. Tetrandrine Ameliorates Myocardial Ischemia Reperfusion Injury through miR-202-5p/TRPV2. Biomed Res Int. 2021 Mar 8;2021:8870674. DOI: 10.1155/2021/8870674.

2. 2.8 Statistical analysis: Line 149. The authors mentioned the data are presented as the mean±standard deviation (SD) in the results. However, none of the results and the figures are displayed with SD instead of Standard Error mean. The authors are requested to change SD to SEM in the statistical analysis section to avoid confusing readers.

Response: Thanks for your suggestion. We have been revised in the revised manuscript.

3. In the Result section:

From lines 183 to 207: The authors explained that bacterial strains are differentially expressed in the I/R model and reverse in Irisin-treated models. However, the authors repeated the bacterial names, which were hard to read and understand, and reviewed the results. The authors are requested to simplify the presentation in an easily understandable manner.

Response: Thanks for your suggestion. In Section 3.2, we described the sequencing results of gut microbiota to show the effect of irisin on the composition of gut microbiota at the phylum, family, genus and species levels. Bacterial names were obtained through the standard database https://www.arb-silva.de/ and could not be stated simply or changed.

4. Figure legend section:

Lines 494-512: Figure 4, Figure 5, and Figure 6: The authors did not explain the subsections' (A-F) self-explanatory information in the legends alphabetically. Comparing what is shown in the figures and the figure legends is hard. The authors are requested either change the subsections alphabetically based on the figure legends or figures accordingly.

Response: Thanks for your suggestion. The legend for Figure 4, Figure 5, and Figure 6 have been corrected in the revised manuscript. In figure 4 (A, B and C), Figure 5 (A, B and C) and Figure 7 (B and C) scale bars have been added clearly on the images.

5. Additional experiments may need to more strongly validate the Irisin mediated reversal of barrier integrity findings. The authors used only LPS levels for blood parameters to validate the barrier dysfunction. The authors are requested to consider to do to measure Zonulin and FITC-Dextran levels in the blood in both experimental models, which will strengthen the manuscript.

Response: Thank you for your guidance and making the article more complete. The results of Zonulin levels in the blood have been supplemented in the revised manuscript (Fig 6A). In general, rats are gavaged with FITC dextran for 4 hours before collecting peripheral blood for testing intestinal permeability. In acute animal models, animals have a short survival time, so the FITC-Dextran assay may not be appropriate.

---

## [Decision Letter · Decision Letter 1]

21 Aug 2023

Irisin ameliorates myocardial ischemia-reperfusion injury by modulating gut microbiota and intestinal permeability in rats

PONE-D-23-09095R1

Dear Dr. Fan,

We’re pleased to inform you that your manuscript has been judged scientifically suitable for publication and will be formally accepted for publication once it meets all outstanding technical requirements.

Kind regards,

Pradeep Dudeja

Academic Editor

PLOS ONE

Additional Editor Comments (optional):

Reviewers' comments:

Reviewer's Responses to Questions

**Comments to the Author**

1. If the authors have adequately addressed your comments raised in a previous round of review and you feel that this manuscript is now acceptable for publication, you may indicate that here to bypass the “Comments to the Author” section, enter your conflict of interest statement in the “Confidential to Editor” section, and submit your "Accept" recommendation.

Reviewer #1: All comments have been addressed

Reviewer #2: All comments have been addressed

2. Is the manuscript technically sound, and do the data support the conclusions?

Reviewer #1: Yes

Reviewer #2: Yes

3. Has the statistical analysis been performed appropriately and rigorously? 

Reviewer #1: Yes

Reviewer #2: Yes

4. Have the authors made all data underlying the findings in their manuscript fully available?

Reviewer #1: Yes

Reviewer #2: Yes

5. Is the manuscript presented in an intelligible fashion and written in standard English?

Reviewer #1: Yes

Reviewer #2: Yes

6. Review Comments to the Author

Reviewer #1: The authors have provided a nicely detailed and thorough response to the comments from the previous review and have addressed all my concerns.

However, there are following 2 minor mistakes that can be taken care of during proof reading:

1. In line 198 and 199, repetition of "Staphylococcus" is observed.

2. Authors should follow the same format (either uppercase or lowercase) for mentioning "DOI". As in majority of references DOI is written in uppercase but in some reference its mentioned in lowercase (Ref 24, 25, 40, 41, 46, 47 and 48).

Reviewer #2: I appreciated the authors response of each comments raised by the reviewers. This manuscript now look scientifically flawless.

7. PLOS authors have the option to publish the peer review history of their article (what does this mean?). If published, this will include your full peer review and any attached files.

Reviewer #1: No

Reviewer #2: No

---

## [Editor Report · Acceptance letter]

25 Aug 2023

PONE-D-23-09095R1 

Irisin ameliorates myocardial ischemia-reperfusion injury by modulating gut microbiota and intestinal permeability in rats 

Dear Dr. Fan:

I'm pleased to inform you that your manuscript has been deemed suitable for publication in PLOS ONE. Congratulations! Your manuscript is now with our production department. 

Kind regards, 

on behalf of

Dr. Pradeep Dudeja 

Academic Editor

PLOS ONE